# Synchronous transmissions on Bluetooth 5 and IEEE 802.15.4 – A replication study

Romain Jacob, Anna-Brit Schaper, Andreas Biri, Reto da Forno, Lothar Thiele
jacobr@ethz.ch
ETH Zurich

## ABSTRACT

Synchronous transmissions (*ST*) is a wireless communication technique that has been shown to be particularly efficient in low-power multi-hop networks. Since 2011, research on *ST* mainly focused on the physical layer defined by the IEEE 802.15.4 standard. Nowadays, Bluetooth is another pervasive technology embedded by default in almost all connected objects; researchers recently started to investigate whether the benefits of *ST* also apply to Bluetooth.

This paper presents the results of a replication study of *ST* using the popular and low-cost nRF52840 Dongle, which supports all modes of the Bluetooth 5 standard as well as IEEE 802.15.4. We measure the packet reception rate for different parameters known to affect *ST* for all physical layers supported by the platform. We use a data exploration application that allows to extract useful information from the measurements and uncover new insights. We confirm that *ST* is viable on Bluetooth, as previously shown. Moreover, our data show that successful *ST* on Bluetooth cannot be explained by "constructive interference" or capture effect alone: multiple effects interplay in a way that is not yet fully understood.

**Data Availability Statement.** The authors commit to keep all data presented in this paper publicly available for at least 3 years. The dataset and the visualization application code are hosted on GitHub and archived on Zenodo [8].

## 1 INTRODUCTION

Synchronous transmissions (*ST*) (also referred to as concurrent transmissions) is a wireless communication technique that let multiple nodes transmit packets at the "same time." A destination node may successfully receive (one of these) synchronous transmissions thanks to two artifacts of the physical layer: constructive interference and the capture effect. In a nutshell, *ST* is likely to be successful if the incoming messages arrive at the receiving node's antenna within a small time offset (in the range of a few  $\mu$s) and/or with a sufficiently large difference in signal strength (a few  dB). In 2011, Glossy [7] was the first protocol using *ST* for fast and reliable communication over a low-power multi-hop wireless network, using a flooding strategy. This triggered a decade of research, mainly focused on the IEEE 802.15.4 standard. Refer to [14] for more details on *ST* and the associated literature.

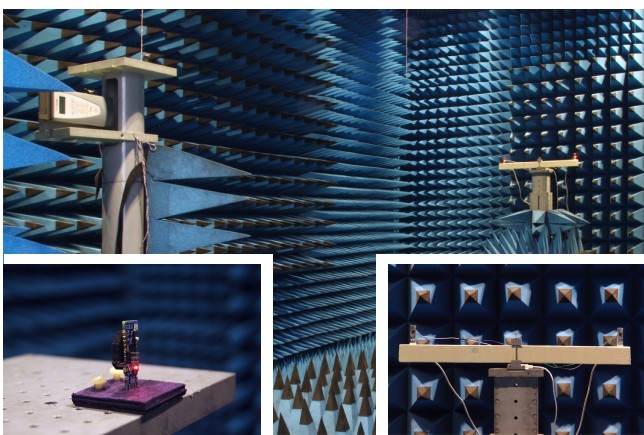

**Figure 1: Experimental setup of the nRF52840 Dongles in an anechoic chamber.** *Left corner: receiver. Right corner: transmitters.*

In 2019, Al-Nahas et al. showed that *ST* can also be successfully used on Bluetooth's physical layer [1]. The authors presented a first characterization of the conditions under which *ST* can be received and presented BlueFlood, a Glossy-like communication protocol where nodes efficiently exchange Bluetooth-compatible advertisement packets. While these first results were promising, some gaps remain in our understanding of how *ST* work on Bluetooth. In particular, while we have an intuition of the parameters that affect the success of *ST*, previous results suggest cross-dependencies between these effects [1], which are yet to be fully characterized. We therefore ask ourselves the following question:

*Can we replicate the results from [1] and confirm the conditions under which ST can be successful on Bluetooth's and IEEE 802.15.4's physical layers?*

To answer that question, we conduct an experimental campaign using the nRF52840 Dongle [10] which is capable of both Bluetooth 5 and IEEE 802.15.4. We focus on the link layer and measure the packet reception rate for different parameters known to affect *ST* for all physical layers supported by the platform. We perform all experiments with two synchronous transmitters set in an anechoic chamber to avoid external interference (Figure 1).

Note that we are investigating the success rate of a *single ST attempt*, which is the elementary component of flooding-based protocols such as Glossy or BlueFlood. These protocols achieve near-perfect reliability by leveraging the spatial and temporal redundancy embedded in the flooding logic, effectively providing multiple *ST* attempts for each packet. In this work, we focus on understanding the mechanisms underlying the success of single *ST* attempts, which ultimately allows to design better-performing protocols (*e.g.*, by increasing the preamble size to tolerate larger transmission time offsets in high-interference scenarios [6]).

**Table 1: Summary of relevant PHY properties for *ST*.**

| IEEE 802.15.4 Modulation | Bitrate [bps] | Chip rate [ps] | Chip period [us] | FEC |
|---|---|---|---|---|
| O-QPSK | 250k | 2M | 0.5 | 1:8 |

| Bluetooth 5 Modulation | Bitrate [bps] | Symbol rate [ps] | Symbol period [us] | FEC |
|---|---|---|---|---|
| GFSK | 2M | 2M | 0.5 | – |
| GFSK | 1M | 1M | 1 | – |
| GFSK | 500k | 1M | 1 | 1:2 |
| GFSK | 125k | 1M | 1 | 1:8 |

## 2 BACKGROUND

This study aims to investigate and compare the conditions for successful synchronous transmissions (*ST*) for the different physical layers (PHY) supported by the nRF52840 platform. Before presenting our results, this section briefly presents the platform (Sec. 2.1), summarizes the relevant properties of supported PHY (Sec. 2.2), and lists the different parameters known to affect *ST* (Sec. 2.3).

### 2.1 The nRF52840 platform

We perform all our experiments using the Nordic Semiconductor nRF52840 Dongle (also known as PCA10059) [10]. The dongle embeds a PCB antenna, a few peripherals, and the nRF52840 system-on-chip [9] including an ARM Cortex-M4, 256 kB of RAM and 1 MB of flash. The dongle is about 1.5 cm × 4.6 cm-large and costs around 10 \$ as of today: it is a cheap and small commercial-off-the-shelf embedded platforms, suitable for all sorts of deployments.

The nRF52840 SoC features some hardware support particularly interesting for *ST*. The *programmable peripheral interconnect* (PPI) allows peripherals to communicate with each other independently of the CPU. The PPI signals are synchronized to a 16 MHz clock, thus have a predictable delay of (at most) 62.5 ns. Certain peripherals also provide so-called *shortcuts*, which are connections between events and tasks within a peripheral. For example, one can use shortcuts to automatically stop or clear a timer when it has reached a user-defined value, then triggers a radio transmission using the PPI; all without involvement of the CPU and thus predictable timing. Finally, the nRF52840 supports multiple radio physical layers, which are further described in the following section.

### 2.2 Physical layers

The nRF52840 plaform supports five different PHY: IEEE 802.15.4 and the four modes specified in Bluetooth 5 [13], which all operate in the 2.4 GHz ISM band. The parameters of these PHY that are relevant for our study of *ST* are summarized in Table 1.

The IEEE 802.15.4 standard uses O-QPSK modulation with DSSS forward error correction (FEC), encoding 4 bits of data into a symbol made of 32 chips (a chip is an analog 0 or 1). The chip rate is 2 Mbps which yields a chip period of 0.5 μs and a data bitrate of 250 kbps.

The Bluetooth 5 standard describes 4 modes, all using GFSK modulation. They are best identified by their bitrate of 2 Mbps, 1 Mbps, 500 kbps, and 125 kbps respectively. These modes use different symbol rate; the slowest two use convolution coding for FEC with 1:2 and 1:8 rate (see *e.g.*, [2] for more details).

### 2.3 Parameters affecting the reception of *ST*

*ST* refers to a situation where multiple transmitters in range of the same receiver simultaneously send packets. *ST* is considered successful if the receiver correctly decodes one of the transmitted packets. The PHY supported by the nRF52840 (Sec. 2.2) are based on phase and frequency modulation for which the following parameters are known to affect the success of *ST*.

**Power delta** By design of RF receivers based on frequency modulation, if one signal is sufficiently stronger than other interfering signals and still arrives during the preamble of previous transmissions, the receiver locks onto the stronger signal and decodes the corresponding packets with high probability. This is known as the *capture effect* and requires a power difference of 3 dB to 10 dB depending on the PHY.

**Packet content** In an ideal scenario, the signal from different transmitters arrive at the receiver with the same phase offset. If the packets are the same, this can lead to constructive interference, resulting in a strictly stronger signal.

**Time delta** Even if the packets are the same, the received signals invariably have an offset in the time domain. If this offset is larger than the symbol period $\tau$, the received symbols superpose randomly and reception fails (assuming no capture effect). However, if the time offset is small ($\tau$ or less), it has been experimentally shown that successful *ST* is likely. This effect is referred to as *"constructive interference."*

**Coding** Some PHY use coding mechanisms to improve the reliability of transmissions, which also affects the reception *ST*.

**Carrier frequency offsets** Transmitters do not have the exact same carrier frequency, which is particularly true for cheap commercial-off-the-shelf platforms. With two transmitters, the envelop of received signal has a sinusoidal shape, which creates time windows with stronger and weaker signals. This is known as the beating effect.

**Environment** Naturally, interference from other radio sources affects *ST*. Moreover, reflections of the transmitted signals may also reach the receiver (multipath effects), which creates additional signals with other phases and may affect the reception of *ST* in peculiar ways.

**Number of transmitters** With more transmitters, all the previous effects add up and are mixed together in ways that are difficult to predict. In an ideal scenario, more transmitters could perfectly superimpose and produce a stronger signal; the reality is more complex and less predictable.

## 3 EXPERIMENT DESIGN

Our experimental campaign aims to investigate the success rate of a single *ST* attempt with two synchronous transmitters while varying four variables: the physical layer used, whether the transmitted packets have the same content or not, and the time and power delta between the signals at the receiver. This section briefly presents our experiment design; refer to [11] for more details.

**Setup.** Our physical setup is illustrated in Fig. 2: two transmitters are placed at equal distance of the receiver, such that we can control the received signals time delta with the time offset at the transmitters. We run the experiments in an anechoic chamber to minimize external interference and signal reflections (Fig. 1).

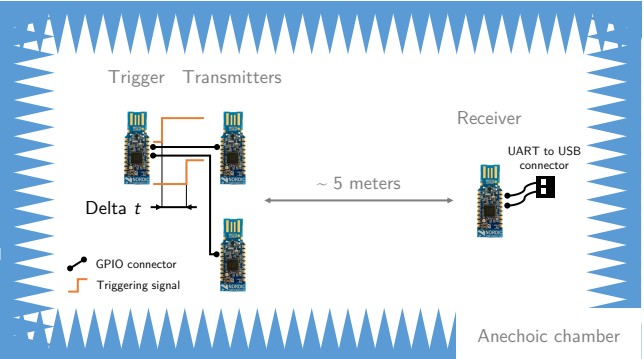

**Figure 2: Schema of the experimental setup** *(pictured in Fig. 1).*

**Statistical methods.** We aim to estimate the reliability of *ST* using a sound statistical analysis, which requires independent data samples [3]. For such estimations, performing many attempts in a row is not useful, as they would be all correlated; instead, one must perform independent runs of the same experiment, compute the metrics of interest for each run, then estimate the expected value of the metrics based on the runs' data. This favors a experiment design with short runs (performing only a few *ST* attempts per settings) and many repetitions of these runs.

**Runs.** Experiments are performed in runs where multiple parameter settings are tested in sequence. For each run and each setting, 20 *ST* are performed: the receiver logs the number of successful attempts, which is our reliability metric. Runs are repeated at least 5 times, which allows to compute a 75% confidence interval of the median reliability [3]. We intentionally trade few repetitions for a large number of different settings (see **Parameters**).

The runs are time-triggered and controlled by the trigger node on the transmitting side (Fig. 2). The sequence of settings is predefined such that nodes know how to set their radio correctly. Finally, the packet payload contains the current setting to guarantee that the receiver logs the correct data, even in case of packet losses.

**Time delta setting.** The time delta between transmitters is controlled by the trigger node; it raises GPIO pins connected to the transmitters with a precise offset, which trigger the transmissions. We analyzed the jitter on the resulting time delta between transmissions, which results in an accuracy of 124 ns (*i.e.*, ≈ 2 ticks) [11].

**Power delta setting.** While the radio can be configured with different transmit power settings, it is not clear *(i)* how precise these settings are, and *(ii)* what the actual signal strength at the receiver is. Therefore, we chose to start each run with a series of RSSI measurements. For each setting, each transmitter sends 20 packets for which the receiver measures and logs the corresponding RSSI value. These measurements are used in post-processing to estimate the actual received power difference between the transmitters in each of the different settings. According to the nRF52840 datasheet, the RSSI measurements have an accuracy of ±2 dB.

**Parameters.** We chose to test the following settings.

- We test the five physical layers supported (see Table 1).
- The time delta set by the triggering node are ± 0..15, 20, 25, 30, 35, 40, 45, 50, 60, 70, 80, 90, 100, 120 ticks.

- One transmitter sets its radio to 8 dBm while the other cycles through -8, -4, 0, 2, 3, 4, 5, 6, 7 and 8 dBm.
- Transmitters send 38-byte Bluetooth-compatible advertisement packets. When packet content should be different, 14 bytes (out of 38) are randomly generated; the others are fixed by Bluetooth or used to store the current settings.

We fix the packet content parameter in each run and cycle through the other settings; this yields 5900 combinations and 118'000 *ST* attempts per run, resulting in about 74 min of runtime.

**Data collection.** For each of the 5900 setting, 20 *ST* attempts are performed. The receiver logs the number of successes, which fits in one byte. In addition, the receiver must log the RSSI measurement: 10 power settings, 5 modes, 2 transmitters, and 20 attempts result in 2000 measurements of one byte each. Thus one run produces 7.9 MB of data, which easily fits in the nRF52840 memory. Once the run is finished, the receiver is connected to a laptop; the RSSI and *ST* data are written over UART and finally stored as CSV files.

## 4 RESULTS

The main goal of our study is to *(i)* measure the tolerable time delay for "constructive interference" and the power delta threshold for the capture effect and *(ii)* compare these to previous studies [2, 12]; This goal is fulfilled by Table 2. The following figures provide more details: Fig. 3 illustrates that the capture effects works on all modes while Fig. 4 presents the data for the "constructive interference" effect. Finally, Fig. 5 shows that there is a region where successful *ST* cannot be explained by solely "constructive interference" or the capture effect.

**Plotted data.** In Fig. 3 to 5, the dots (when displayed) show the packet reception rate (PRR) for one run, which is computed over 20 *ST* attempts (Sec. 3). Based on all the runs, we compute, for each setting, the median PRR (solid line) and its 75% confidence interval (shaded areas); *i.e.*, given the collected data, there is a 75% probability that the true median PRR is within the confidence interval. We use TriScale [3] to compute the confidence intervals.

**Threshold definitions.** In Table 2, we report the thresholds we observe for "constructive interference" and the capture effect. We consider "constructive interference" when the confidence interval of the median PRR raises above 0. For the capture effect threshold, we take the minimal power delta such that the confidence interval of the median PRR is above 75% for all time delta. Note that previous studies [2] did not formalize their threshold definitions, nor released their dataset, which makes comparison and replication difficult.

**Findings.** Overall, we obtain three main findings:

(1) We confirm that *ST* is viable on all Bluetooth physical layers included in the Bluetooth 5 standard [13] as well as IEEE 802.15.4. When the same packet is sent by the transmitters (Fig. 3a), the median PRR is close to or larger than 50% even without any power delta. In these conditions, the "constructive interference" effect helps the reception of *ST*. When the packets sent are different (Fig. 3b), the PRR requires a larger power delta (between 2 and 10 dB depending on the mode) to reach 100%.

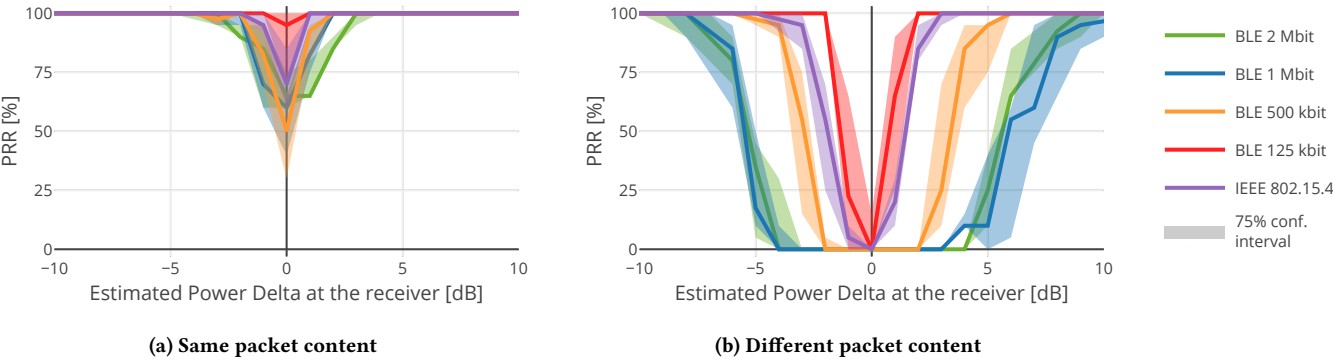

(a) Same packet content

(b) Different packet content

**Figure 3: [Measurements with $0\,\mu s$ time delta between the transmitters]** *ST* **is successful for all modes when the power delta at the receiver becomes sufficient.** *When the same packet is sent by the transmitters (Fig. 3a), the median PRR is close to or larger than 50% even without any power delta. In these conditions, the "constructive interference" effect helps the reception of ST. When different packets are sent (Fig. 3b), the PRR requires a larger power delta (between 2 and 10 dB depending on the mode) to reach 100%. The minimum power delta beyond which ST is successful independently of the time delta (i.e., capture effect threshold) is even larger (see Fig. 5). Overall, these results match those presented in [2].*

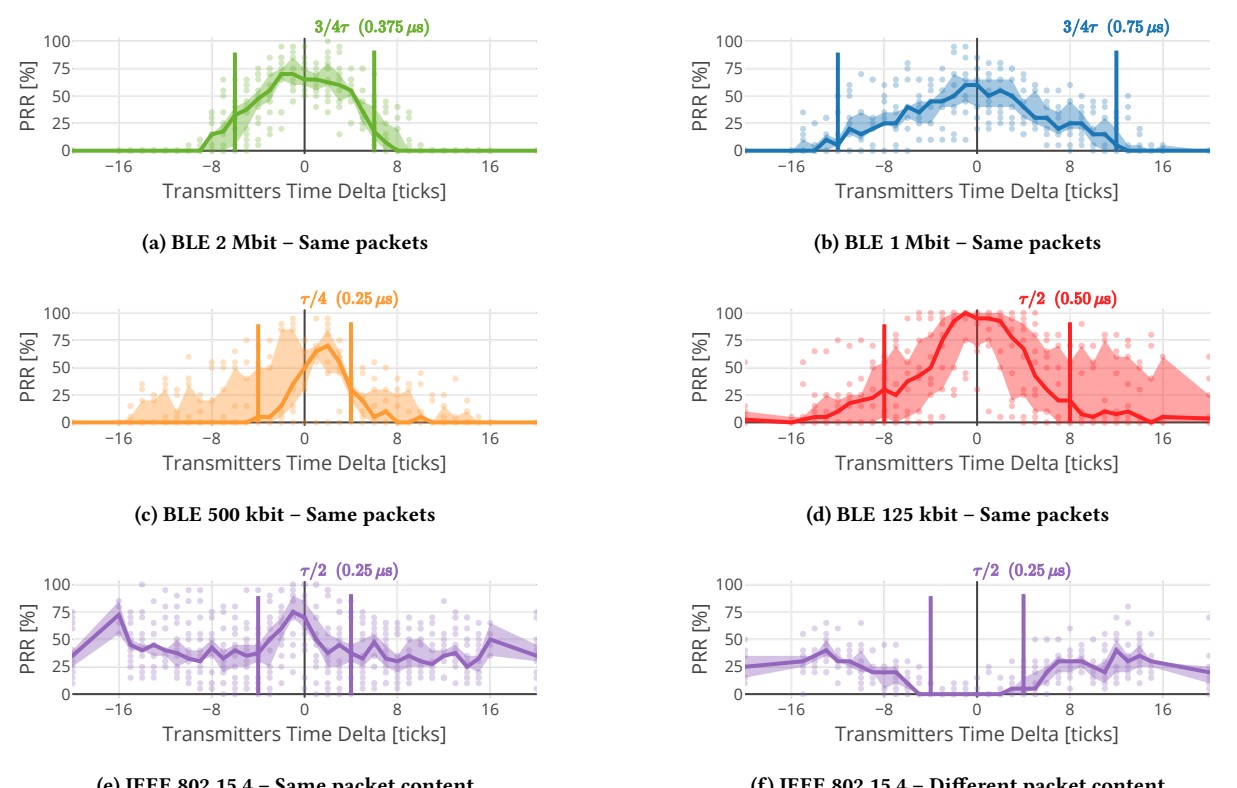

(a) BLE 2 Mbit – Same packets

(b) BLE 1 Mbit – Same packets

(c) BLE 500 kbit – Same packets

(d) BLE 125 kbit – Same packets

(e) IEEE 802.15.4 – Same packet content

(f) IEEE 802.15.4 – Different packet content

**Figure 4: [Measurements with 0 dB received signal difference at the receiver (estimated)]** **Without power delta, *ST* can still be successful when *(i)* the same packets are sent and *(ii)* the time delta between the transmitters are sufficiently small. This is the so-called "constructive interference" effect.** *For the 4 Bluetooth modes (Fig. 4a to 4d) the median PRR drops to 0 when the time delta between transmitters becomes too big. The bounds found in our experiments are marked on the graph and labeled with the tolerable time delta (in ratio of the symbol period and the corresponding time in $\mu s$). In [2], the authors conclude that Bluetooth modes cannot tolerate more than $\tau/4$ of delay; our results show that some modes can. For IEEE 802.15.4, the PRR never drops to 0 thanks to the DSSS error correction (Fig. 4e); thus we redefine the "constructive interference" region as the time deltas for which the PRR is 0 when transmitters send different packets (Fig. 4f). We observe a limit around $\tau/2$ (or 0.25 $\mu s$), which matches previous studies [5, 12].*

**Table 2: Conditions for capture effect and "constructive interference" reported in previous studies, compared with our measurements (in bold).** *We find that Bluetooth modes can tolerate larger time delay than concluded in [2] while still benefiting from "constructive interference." Conversely, our threshold for capture effect are slightly more conservative. However, these differences are moderate and could be simply explained by the different definition of these time and power thresholds (not clearly defined in [2]). $\tau$ denotes the symbol or chip period for the different PHY (see Table 1); consequently, $\tau = 0.5\,\mu s$ for IEEE 802.15.4 and BLE 2 Mbit, and $\tau = 1\,\mu s$ for the other Bluetooth modes.*

|  |  | Study | IEEE 802.15.4 | BLE 2 Mbit | BLE 1 Mbit | BLE 500 kbit | BLE 125 kbit |
|---|---|---|---|---|---|---|---|
| *Power difference threshold for the capture effect* | | | | | | | |
| Conditions | Unit | [12] | 3 | 10 | 10 | – | – |
| - any time delta | | [2] | – | 8 | 8 | 8 | 2 |
| - any payload | dB | **This study** | **3** | **10** | **10** | **8** | **6** |
| *Tolerable delay for constructive interfence* | | | | | | | |
| Conditions | Unit | [12] | $\tau/2$ (0.25) | $\tau/2$ (0.25) | $\tau/2$ (0.5) | – | – |
| - 0 dB delta | | [2] | – | $\tau/4$ (0.13) | $\tau/4$ (0.25) | $\tau/4$ (0.25) | $\tau/4$ (0.25) |
| - same payload | $\tau$ ($\mu$s) | **This study** | **$\tau/2$ (0.25)** | **$3/4\tau$ (0.375)** | **$3/4\tau$ (0.75)** | **$\tau/4$ (0.25)** | **$\tau/2$ (0.5)** |

(2) We measure the packet reception rate (PRR) of *ST* on the nRF52840 Dongles; the capture effect thresholds and tolerable delays for "constructive interference" that we obtain (Table 2) slightly defer from previous studies [2, 12]. These difference may be due to the inherent imprecision of the time and power measurements, as well as the lack of the formal definition of these threshold in the previous studies, which makes comparisons difficult.

(3) We confirm the evidence of a region where good PRR is observed but can not solely be explain by capture effect or constructive interference (Fig. 5): a power delta smaller than the capture threshold (Table 2) still improves the reception of *ST* for moderate time delta. In other words, even a small power delta increases the tolerable time delay and improves the PRR.

The raw data, processing scripts, and data visualization are all publicly available on GitHub and archived on Zenodo [8]. The data visualization app can be run directly online at explore-st-data.ethz.ch.

## 5  LESSONS LEARNED

During this study, we learned a few lessons we deem worth sharing.

- Fig. 5 clearly shows that *ST* can be successful on the Bluetooth physical layers under conditions that can neither be attributed to the capture effect nor "constructive inference". Success may depend on other parameters (coding scheme, radio transceiver design, etc.); this is not yet fully understood.
- The carrier frequency offset between transmitters is an important parameter which leads to the well-known "beating effect" [5]. It has been shown experimentally that the beating frequency strongly affects the performance of *ST* [2]. It is not clear whether imprecise and unstable oscillators of cheap commercial-off-the-self platforms like the nRF52840 Dongle are detrimental or beneficial with respect to the beating effect. In our study, we tried different devices but obtained only mildly different carrier frequency offsets, leading to minor effects.

- A careful study of *ST* implies working with a lot of data with many dimensions. We found that having an efficient data visualization framework is not just nice to have, it is *necessary*. We make our visualization tools available hoping they can be useful for others too.
- The timing of operation executions on the nRF52840 can be made very predictable thanks to built-in hardware support (PPI and shortcuts [9]). This significantly facilitates the design of communication protocols based on *ST*; in particular, achieving the sub-$\mu$s time synchronization accuracy necessary to benefit from "constructive interference" is now much easier than it used to be (*e.g.*, for the initial Glossy [7]).
- The receptive field of the PCB antenna on the nRF52840 Dongle is far from perfect. During our experimental campaign, we sometimes observed significant differences between runs after touching the boards. Such effects cannot be observed when using coaxial cables, like in [2].
- As mentioned in Sec. 3, the precision of time (±2 ticks) and power delta (±2 dB) is in the order of the threshold we try to identify. The results in Table 2 must be taken with care. Note that previous studies suffer from similar imprecision, which are hardly avoidable.

## 6  CONCLUSIONS AND FUTURE WORK

With this study, we confirm that synchronous transmissions (*ST*) is viable on the physical layers of Bluetooth 5. We confirm that *ST* works reliably even on cheap commercial-off-the-shelf platforms like the nRF52840 [10]. The data we collected sheds more light on the expected performance of *ST* for various time and power delta between the signals of two transmitting devices. In particular, we highlight that small power delta improves the reliability of *ST*, even though the conditions for the capture effect are not met (Fig. 5).

Yet, there are still a lot we do not fully understand. In [2], the authors attempted to model the effect of beating on *ST* for the different Bluetooth modes, and validated their theory with experiments with two transmitters. It is expected that the beating effect "averages out" with more transmitters, which should yield an increase in

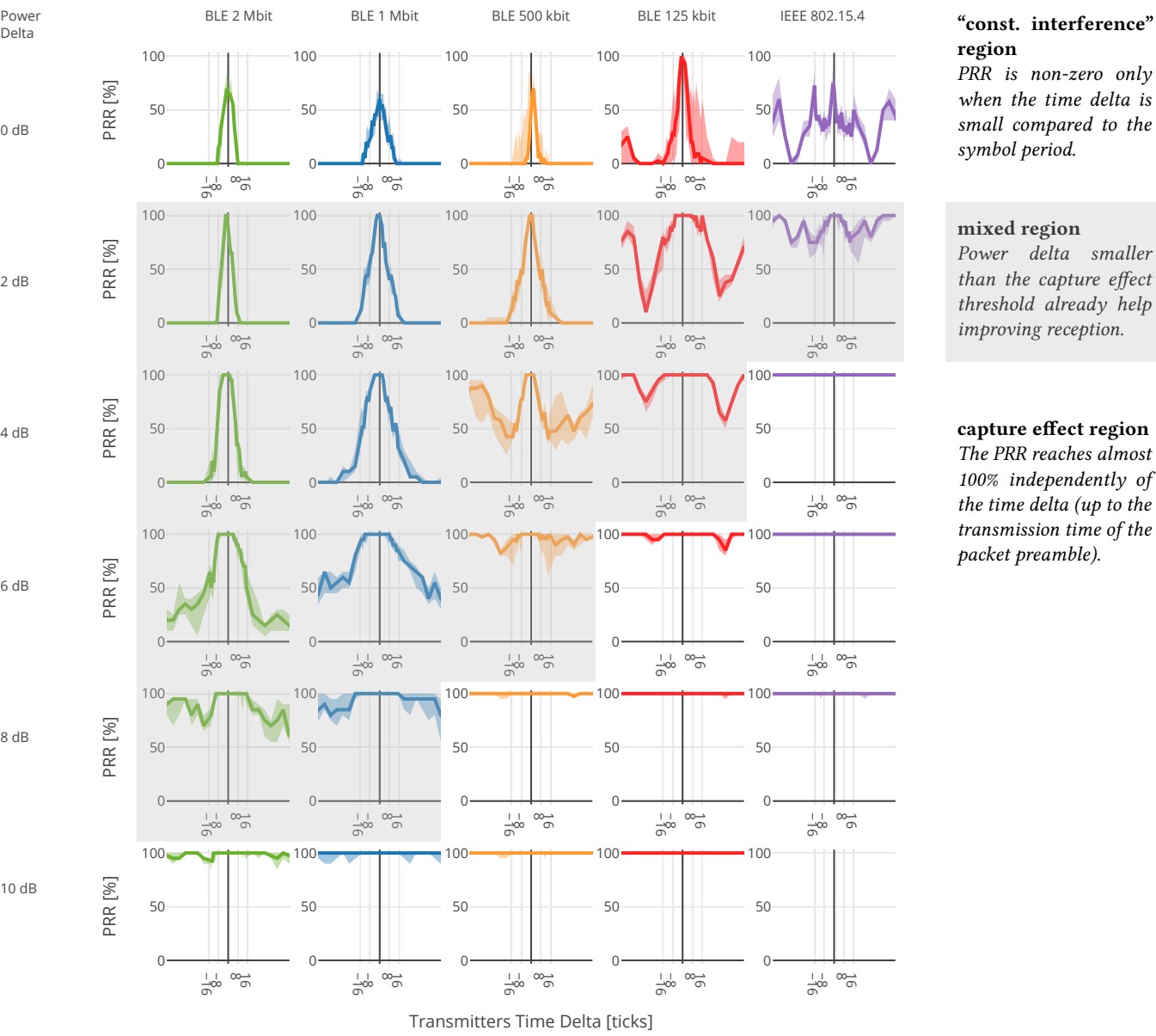

**Figure 5: [Measurements with the same packet content. PRR as a function of the time delta between transmitters for the different modes (columns) and power delta (rows)]** For the Bluetooth modes, it is not all "constructive interference" or "capture effect": a power delta smaller than the capture threshold (Table 2) still improves the reception of *ST* for moderate time delta. In other words, even a small power delta increases the tolerable time delay and improves the PRR. *For example, consider the 1 Mbit mode: we observe a capture threshold (i.e., when ST becomes successful regardless of the time delta) at about 10 dB. However, with only 6 dB power delta, the median PRR is close to 100% for time delta below 16 ticks (1 µs). A similar observation can be made for the 125 kbit mode: a 2 dB power delta is sufficient to provide good reliability up to 8 ticks time delta (0.5 µs).*

performance for the uncoded modes (1 Mbit and 2 Mbit). The effects are actually more complex as shown in a concurrent study [4].

Most importantly, apart BlueFlood, introduced in [1] as a proof-of-concept, the design on multi-hop communication protocols based on *ST* using Bluetooth is largely unexplored. The different Bluetooth modes offer a broad design space in reliability, bandwidth, and range properties: How to leverage these to improve network-wide performance? How does this compare with the performance of Bluetooth Mesh (the multi-hop protocol included in the Bluetooth 5 standard)? How does it compare with existing solutions using the IEEE 802.15.4 physical layer? Again, Baddeley et al. [4] provides some answers, but the question remains generally open.

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
