# OpenReview forum: "Synchronous transmissions on Bluetooth 5 and IEEE 802.15.4 – A replication study"
_sigmobile.org/MobiCom/2020/Workshop/CPS-IoTBench — CPS-IoTBench 2020 Conditional_

### Official Review · AnonReviewer2 · 2020-06-26
**interesting paper but lacks on presentation**

**Rating:** 6
**Confidence:** 4

**Review:**

This paper explores the area of synchronous transmissions (ST) and the effect of constructive interference on the packet received ration (PRR). ST is a challenging topic where many variables affect the outcome. In this paper authors are trying to systematically study several variables and their effect on PRR. The authors evaluate the ST on two types of radio - IEEE 802.15.4 and Bluetooth. In the case of Bluetooth, the authors evaluate 4 different modes.

The proposed test-bed and evaluation technique seem to be valid and sound.

While the paper is explores a very interesting area and performs thorough study of various parameters affecting ST, the presentation of the result is somewhat short and left on the reader to infer some of the results. I think, the paper would benefit if the figures were made smaller and a text describing the results in more details would be added.

Also, it is not clear why authors used only 20 packets to evaluate PRR. This number is rather small to draw final conclusions. I would recommend to repeat the experiments with at least 100 packets sent.

This paper explores the problem of synchronous transmissions in the case of various types of Bluetooth radios and study the effect of various variables (time delta, power delta, and same/different packet) on the final PRR. The results showed in the paper are a valuable contribution in this area of research, however, the presentation of the paper can be improved.

---

### Official Review · AnonReviewer1 · 2020-06-28
**not much novelty, not enough PRR samples, weird results**

**Rating:** 3
**Confidence:** 5

**Review:**

Positives:

- synchronous transmission are a hot topic, whose mechanisms have been
  applied more than understood
- considers two radios
- interesting setting in anechoic chamber

Negatives:

- 20 transmissions for each experiment is not enough to assess PRR
- results for 802.15.4 conflict severely with previous knowledge
- few novel findings, if any

This paper aims at characterize synchronous transmissions for both
IEEE 802.15.4 and BLE, using a platform that supports both. In
principle, this is a commendable effort, especially given the setup
(anechoic chamber) and the many parameters under test.

On the other hand, in my opinion this paper has a fundamental flaw,
namely, for each combination of parameters only 20 synchronous
transmissions (ST) are considered. This number is way too small to be
statistically relevant: a single failure yields immediately a drop to
PRR=95%. This biases the results in the paper (i.e., the main
contribution) in an acceptable and irremediable way.

Furthermore, some of the results obtained conflict with the large body
of literature accrued on the topic. Fig.4e and the IEEE 802.15.4
results in Fig.5 are not what one would expect when comparing the
results at 0dBm obtained with the CC2420, CC430, CC2538, which yield
much higher reliability.

Other minor points:

- first sentence of the intro: I wouldn't characterize ST as "a
  wireless communication technique that allows multiple nodes to
  transmit a message at the “same time.” That's actually not the
  point. The point is actually the reverse: if messages are
  transmitted at the "same time" they at least one of them is much
  more likely to be received.

- in the time delta description: capture is also involved, not just
  constructive interference

- I didn't get the meaning of the thresholds, and the assignment to the type of phenomenon observed, which appears quite arbitrary to me

- it would be much better to report time instead of ticks

---

### Official Review · AnonReviewer3 · 2020-06-28
**Interesting study but needs more depth**

**Rating:** 3
**Confidence:** 5

**Review:**

Summary:
Synchronous transmissions (ST) have garnered attention as they offer low-latency flooding/synchronization/data collection protocols, primarily over 802.15.4 networks. BlueFlood showed that ST can work on a BLE mesh. This articles attempts to investigate this, however falls short.

Strengths:
* Controlled experiments anechoic chamer
* Identifies an easier way for getting precise timing on nodes

Weaknesses:
* Of the 3 questions raised in the introduction, only the first two are interesting (3rd cannot be treated as a separate one since BlueFlood demonstrated on COTS nrf52840 devices).
* Limited experimentation - number of nodes and runs are too limited to conclude on the working of ST or CI.
* Results are not matching with literature. Glossy shows close to 100% PRR in 802.15.4  using CC2420 nodes
* Results lack explanation. The reader is left with insufficient details.
* Table 2 presents interesting numbers over [2] and [10]. Any theoretical backing or reason why the tolerable delays are more?
* Figure 4 is leaves one to wonder. Why are the peak PRRs are to the left of '0'? Why is it different in 4(c)? This creates doubts on the timing and experimentation.
* The second question raised is unanswered. Why raise it at all?

Other points:
* Experimentation with SDR may yield more results. See [Wilhelm et al.](https://ieeexplore.ieee.org/abstract/document/6880859/)
* You can use a spectrum analyzer to verify the tx power of each node as this is a controlled experiment setting
* Figure 5 caption is vague

---

### Official Review · AnonReviewer4 · 2020-06-29
**Nice baseline study for synchronous BLE transmissions, just wish distance was considered!**

**Rating:** 7
**Confidence:** 3

**Review:**

This paper presents a systematic evaluation of synchronous transmission with commodity BLE radios. The analysis sweeps transmission power, timing offset, and data rate, and evaluates when the capture effect and constructive interference occur.

Quantifying the effect of synchronous transmissions in BLE establishes a useful baseline for work that will build on this primitive using BLE radios. The study in an anechoic chamber enables repeatable results. The paper also uses standard BLE radios that many designs use and will use in the future.

What surprised me about this paper is that the authors did not consider distance in their study. When thinking about BLE, and how it is commonly used in mobile/wearable devices, distance between the transmitter and receiver seems like a very important and relevant variable. With 802.15.4, one might be able to argue the devices are static with some perhaps known inter-device ranges, but with BLE that argument would be a lot harder to make. Sweeping BLE bitrates is probably less important than understanding how differences in TX-RX distance affect synchronous transmissions.

The introduction could provide more of a description on the envisioned use cases for synchronous transmissions that the authors are using when defining the experimental parameters. Are all nodes cooperating in a global flood like in Glossy? Are devices operating completely independently? Are devices doing pairwise synchronization and then transmitting? This scope would provide more context for the experiments and relevant variables.

The main unexpected item in this paper is that PRR improved even with high timing offsets and relatively low power differences. Capturing this in a single graph would be very useful, to really show the effect happening.

Other notes:
- Why are 14 bytes random in the non-identical packet case? Where did the number 14 come from?
- I would recommend not making the need for a data visualization framework the first lesson learned since that is not the focus of the paper.

---

### Decision · Program_Chairs · 2020-07-07

Conditional Accept